# Value of Artificial Intelligence in Evaluating Lymph Node Metastases

**DOI:** 10.3390/cancers15092491

**Published:** 2023-04-26

**Authors:** Nicolò Caldonazzi, Paola Chiara Rizzo, Albino Eccher, Ilaria Girolami, Giuseppe Nicolò Fanelli, Antonio Giuseppe Naccarato, Giuseppina Bonizzi, Nicola Fusco, Giulia d’Amati, Aldo Scarpa, Liron Pantanowitz, Stefano Marletta

**Affiliations:** 1Department of Diagnostics and Public Health, Section of Pathology, University of Verona, 37134 Verona, Italy; nicolo.caldonazzi@gmail.com (N.C.); paolachiararizzo@gmail.com (P.C.R.); aldo.scarpa@univr.it (A.S.); stefano.marletta92@gmail.com (S.M.); 2Department of Pathology and Diagnostics, University and Hospital Trust of Verona, 37126 Verona, Italy; 3Department of Pathology, Lehrkrankenhaus der Paracelsus Medizinischen Privatuniversität, Provincial Hospital of Bolzano (SABES-ASDAA), 39100 Bolzano-Bozen, Italy; ilaria.girolami16@gmail.com; 4Division of Pathology, Department of Translational Research, New Technologies in Medicine and Surgery, University of Pisa, 56126 Pisa, Italy; nicolo.fanelli@unipi.it (G.N.F.); giuseppe.naccarato@unipi.it (A.G.N.); 5Division of Pathology, IEO, Europefan Institute of Oncology IRCCS, University of Milan, 20122 Milan, Italy; giuseppina.bonizzi@ieo.it (G.B.); nicola.fusco@ieo.it (N.F.); 6Department of Oncology and Hemato-Oncology, University of Milan, 20122 Milan, Italy; 7Department of Radiology, Oncology and Pathology, Sapienza, University of Rome, 00185 Rome, Italy; giulia.damati@uniroma1.it; 8Department of Pathology, University of Michigan, Ann Arbor, MI 48104, USA; lironp@med.umich.edu; 9Department of Pathology, Pederzoli Hospital, 37019 Peschiera del Garda, Italy

**Keywords:** digital pathology, artificial intelligence, lymph nodes, metastases

## Abstract

**Simple Summary:**

In surgical pathology, the assessment of the presence of lymph node metastases is a key aspect in terms of the staging and prognosis of cancer patients. This type of work is time-consuming and prone to error. Owing to digital pathology, artificial intelligence (AI) applied to whole slide images (WSIs) of lymph nodes can be exploited for the automatic detection of metastatic cells, so this task can be automated and standardized, increasing diagnostic quality. This manuscript aims to systematically review the published literature regarding the application of various artificial intelligence systems for the assessment of metastases in lymph nodes in whole slide images.

**Abstract:**

One of the most relevant prognostic factors in cancer staging is the presence of lymph node (LN) metastasis. Evaluating lymph nodes for the presence of metastatic cancerous cells can be a lengthy, monotonous, and error-prone process. Owing to digital pathology, artificial intelligence (AI) applied to whole slide images (WSIs) of lymph nodes can be exploited for the automatic detection of metastatic tissue. The aim of this study was to review the literature regarding the implementation of AI as a tool for the detection of metastases in LNs in WSIs. A systematic literature search was conducted in PubMed and Embase databases. Studies involving the application of AI techniques to automatically analyze LN status were included. Of 4584 retrieved articles, 23 were included. Relevant articles were labeled into three categories based upon the accuracy of AI in evaluating LNs. Published data overall indicate that the application of AI in detecting LN metastases is promising and can be proficiently employed in daily pathology practice.

## 1. Introduction

The incidence of cancer has been increasing worldwide due to a growing and aging population coupled with the adoption of screening programs [1]. One of the most relevant prognostic factors for cancer patients is the presence of lymph node (LN) metastasis. Metastatic disease is an important feature that impacts patient clinical staging and treatment decisions [2]. However, having pathologists manually review LNs microscopically for the presence of metastatic tumor cells is a tedious, time-consuming, and potentially error-prone process. Additionally, many hospitals require intraoperative examinations of sentinel LNs on frozen sections for guiding surgical procedures (Figure 1) [3]. Currently, pathologists may be required to screen a large number of slides of lymph nodes, often including additional immunohistochemical (IHC) stains to conventional hematoxylin and eosin (H&E)-stained sections. As a result, this has increased the workload for surgical pathologists. 

Recently, whole slide imaging (WSI) has made digital pathology (DP) more useful for primary diagnosis [4,5] and non-clinical purposes (e.g., education and research) [6,7,8]. By transforming glass slides into WSIs, it has now become possible to use computerized digital image analysis systems including artificial intelligence (AI)-based deep learning (DL) algorithms to analyze digital slides [9]. DP has demonstrated a strong performance in various tasks in different fields, including pathology applications, some of which have already been approved by the Food and Drug Administration (FDA) [10]. In recent years, many AI-based algorithms have been created for the automatic detection of metastases in LNs in WSI. Such novel technology exhibits the potential to reduce pathologists’ workload and increase diagnostic accuracy. The aim of this manuscript is to systematically review the published literature regarding the application of various AI systems for the assessment of metastases in LNs in WSIs. In the next sections of the paper, the search strategy, along with the main characteristics of the retrieved studies, are, respectively, reported in the Material and Methods and the Results. Hence, the included papers are further analyzed in specific paragraphs of the Discussion section according to the following fields of application: (i) the value of AI in lymph node metastases of breast cancer, (ii) the results of public challenges employing and comparing different AI tools, (iii) the value of AI in lymph node metastases detection during intraoperative consultation, and (iv) the role of AI in identifying nodal metastases of tumors apart from breast cancer.

## 2. Materials and Methods

A systematic review of the literature was conducted, without language restrictions, according to the guideline for Preferred Reporting Items for Systematic Reviews and Meta-Analysis (PRISMA) [11] and Meta-Analysis of Observational Studies in Epidemiology (MOOSE) [12]. The databases, Pubmed and Embase, were systematically searched until August 2022 to identify any study regarding the application of image analysis and/or AI for the detection of lymph node metastases. The search strategy comprised a combination of terms including “image and analysis”, “artificial and intelligence”, “morphometry”, “histomorphometric”, “neural network”, “convolutional”, “computational”, “deep learning”, “automated”, “machine learning” “lymph node”, “metastasis”, “WSI”, and their spelling variations. The complete search strategy for these databases is detailed in Appendix A. Two authors reviewed all article titles and abstracts with the aid of Rayyan QCRI reference manager web application [13]. Eligibility of published studies was determined independently by two reviewers with disagreement resolved through consensus. Full texts of the articles fulfilling the initial screening criteria were acquired and reviewed. Inclusion criteria encompassed the application of any kind of AI as a tool for automatic detection of metastasis in LNs. Only full texts of the articles fulfilling the initial screening criteria were acquired and reviewed. Any disagreement with respect to inclusion of a particular article was resolved by consensus.

Studies represented only by abstracts were excluded, as well as reviews and published letters to the editor with no original data. Two investigators independently extracted data from the included studies with a standardized form. Data extracted included author(s), publication year, country of origin for the research, type of metastatic cancer, type of AI employed, main results, and limitations of the study. As for the performances of the AI systems, three cut-offs were chosen based on whether the precision, accuracy, sensitivity, specificity, or area under the curve (AUC) of the receiver operating curves (ROC) was higher or lower than 95%. This threshold was chosen because it is equivalent to two standard deviations (σ). A shade of green was assigned according to the results achieved by the algorithm proposed by the authors. As the color gradation intensifies, the accuracy of AI for the identification of LN metastases increases as follows: (i) light green for every parameter <95%, (ii) medium colored green for at least one parameter >95%, and (iii) dark green: all parameters >95%.

## 3. Results

A flow diagram of the screening and exclusion of all the articles is shown in Figure 2.

A total of 4584 records were retrieved and screened, with only 23 suitable articles finally included in the analysis. An overview of the included studies is provided in Table 1 and Table 2. Publication dates range from 2000 to 2022 and were geographically distributed mainly in the USA (5/23, 22%) and Europe (5/23, 22%), followed by Taiwan (3/23, 13%), China (3/23, 13%), Canada (2/23, 9%), Japan (2/23, 9%), India (1/23, 4%), Bangladesh (1/23, 4%), and South Korea (1/23, 4%).

In more than half of the included studies (16/23, 70%), the investigated LNs were from breast cancer series. Nineteen (19/23, 83%) of the included studies relied only on H&E stains, while three (3/23, 13%) of them used only IHC for cytokeratin and one study coupled H&E with deep UV excitation fluorescence microscopy [14]. As for the employed AI technology, it is worth mentioning that three studies (3/23, 13%) employed a combination (ensemble) of DL algorithms, ranging from four to thirty-seven, for improving the evaluation of breast cancer metastases in LNs. For these reasons, the authors divided so-called ‘challenge’ studies in a different table (Table 2). DL-based computational pathology approaches required either manual annotation of regions of interest (ROI) on WSIs in fully supervised settings or large datasets with slide-level labels in a weakly supervised setting. Both methods required a training dataset. Nineteen (19/23, 87%) of the included studies used a fully supervised approach, while the remaining four (4/23, 13%) algorithms utilized weak supervision [15,16,17,18]. Most of the studies (21/23, 91%) had a WSI training set, whereas two (1/23, 4%) [19,20] studies did not employ a training set as the algorithm was directly able to recognize neoplastic cells stained by the IHC cytokeratin assay. A WSI validation (hold-out) data set was available in three of the papers (3/23 13%). The number of slides used for the training set ranged from 36 to 1963. In the weakly supervised setting, the number of slides of the training set was much higher than in algorithms that relied upon pixel-level annotations.

In terms of AI efficiency, 11 (11/20, 55%) algorithms achieved a performance of >95% across all parameters (precision, accuracy, sensitivity, specificity, or area under the curve of the receiver operating curves) with all four of the weekly supervised studies reaching this goal. As for the detection of isolated tumor cells (ITC), the 95% cut-off was almost never reached; rather, for these settings the AUC rates ranged from 0.575 [21] to 0.9228 [18]. Only one study reached 100% detection rate for ITC, but with the cost of 0% specificity [22]. Among the different employed AI systems, three (3/20, 15%) were programmed to recognize IHC cytokeratin stains, while the others analyzed only H&E slides. Twelve of the latter studies (12/20, 60%) used Convolution Neural Network (CNN) algorithms built on pre-existing platforms, including Googlenet, AlexNet-GRU, Resnet, Densenet, MobileNetV2, and LYNA, among others, and five (5/20, 25%) were ex novo designed algorithms. Finally, three of the included studies (3/20, 15%) simultaneously tested different AI algorithms utilizing the same dataset of H&E stained WSIs [14,23,24].

**Table 1 cancers-15-02491-t001:** Characteristics of the included studies.

Author, Year, Country	Organ	N. of WSI *	Stain	AI Employed	Scanner	Main Results	Limitations
Weaver, 2003, USA	BC LNS	NA 100 (20+, 80−)	CKAE1-AE3	NA	ChromaVision Automated Cell Imaging and Medical System	AI-based identification of 19/20 micrometastatic cases	Use of IHC
Clarke, 2011, Canada	BC LNS	36/ 102 (43+, 59−)	CK 8–18, CAM5.2	CAD algorithm	Mirax slide scanner	Sensitivity detection of ITCs, micro-, and macrometastases of 57.5%, 89.5%, and 100%	Use of IHC
Litjens, 2016, Holland	BC LNS	98, (48+, 50−) /(42+, 56−)	H&E	In house CNN	3DHistech Pannoramic 250 Flash II slide scanner	Identification of 90% of all micro- and macrometastases	NA
Valkonen, 2017, Finland	BC LNS	170 (70+, 100) /100 (40+, 60−)	H&E	In house CNN	#	Mean AUC 0.970-0.839	NA
Holten-Rossing, 2017, Denmark	BC LNS	NA /900(139+, 761−)	CK7, CAM5.2 CKAE1-AE3	Visiopharm APP 10104	Hamamatsu NanoZoomer-XR	Sensitivity and specificity of 100% and 68.9%	NA
Campanella, 2019, USA	BC LNS	9864/NA	H&E	Resnet 34	Philips IntelliSite Ultra Fast	AUC of 0.966	Big amount of data for the testing set
Liu, 2018, USA	BC LNS	270(170−, 110+) /129(49+, 80−)	H&E	LYmph Node Assistant, or LYNA	3DHISTECH Pannoramic 250 Flash II; Hamamatsu Aperio	AUC of 99.6%, no influence by artifacts (overfixation, poor staining, and air bubbles)	No ITC slides
Steiner, 2018, USA	BC LNS	70 (24−, 46+) /NA	H&E	LYmph Node Assistant, or LYNA	Leica AT2 system at a resolution of 0.25 µm/pixel	Shorter turn-around times with AI for micrometastases and negative images	NA
Matsumoto, 2019, Japan	Gastric cancer	56 (18−, 38+) 27 (26+, 1−)	H&E, UV excitation fluorescence microscopy	VGG16, INCEPTION V3, Inception ResNet V2	Nanozoomer C9600-02, Hamamatsu Photonics	Mean accuracy in fluorescence patch classification of 97.4%, 98.2%, and 97.9%, respectively.	Technology not available in all laboratories
Pham, 2019, Japan	Lung cancer	233, 10/106	H&E	HALO-based AI (CNN VGG network)	Aperio Scanscope CS2 digital slide scanner	Sensitivity of 100% (macro–micro metastases, and ITC)	Limited setting parameters, low specificity
Pam, 2020, China	ESCC, lung cancer	242(110−,132+) /795(222+,573−)	H&E	DeepLab model V3 with ResNet-5026	NA	Accuracy of 94% and 90% in ESCC and lung cancer	No data about digital acquisition process
Jin, 2020, Canada	BC LNS	#	H&E	ConcatNet	#	AUC of 0.924	NA
Hu, 2021, China	Gastric cancer	594/327	H&E	Xception, DenseNet-121, and fused networks	Leica Aperio Versa	Negative Predictive Value 97.99% in patients given neoadjuvant chemotherapy	Lot of work to classify at pixel level
Chuang, 2021, Taiwan	Colon rectal cancer	1963, 219/1000	H&E	ResNet 50	NanoZoomer S360 with a 40× magnification	AUC of 0.99 and 0.99 with macro- and micrometastases	Worse performance with ITC (AUC 0.78)
Ming, 2021, USA	BC LNS	CAMELEON 16-17 dataset	H&E	Multiple instance learning	#	Average AUC of 0.953 ± 0.029 at ×40 magnification	Big amount of data for the training set
Shahab, 2022, Bangladesh	BC LNS	270/54	H&E	AlexNet-GRU	Kaggle dataset	Accuracy, precision, sensitivity, and specificity of 99.50%, 98.10%, 98.90%, and 97.50	NA
Tang, 2022, China	HNSCC	85 (38+,47−) /50 (21+, 29−)	H&E	GoogLeNet, MobileNet-v2, ResNet50, and ResNet101	Pannoramic MIDI, 3DHISTECH Ltd.	Overall accuracy, sensitivity, and specificity of 86%, 100%, and 75.9%	NA
Vulli, 2022, India	BC LNS	CAMELEON 16 dataset	H&E	Fine-tuned DenseNet 169	#	Overall accuracy >97.4%	No ITC slides
Khalil, 2022, Taiwan	BC LNS	68 (54+, 18−)/26 (12+, 14−)	H&E, CK AE1-AE3	Modified FCN based on Shelhamer model	3DHISTECH Pannoramic	Including ITC overall precision 89.6%, recall 83.8%, F1-score 84.4%, and mIoU 74.9%	NA
Huang, 2022, Taiwan	Gastric cancer	983, 110/201	H&E	ResNet50 architecture	NanoZoomer S360 digital slide	AUC of 0.9936, improvement of ITC and micrometastasis identification in a shorter turn-around time (−31.5%, *p* < 0.001)	Big amount of data for the training set

* testing/training. # data from references [25] and [26]. Abbreviations: AI: artificial intelligence, BC: breast cancer, NA: not available, LNS: sentinel lymph node, IHC: immunohistochemistry, ITC: isolated tumors cells, CNN: convolutional neural network, H&E: hematoxylin and eosin, AUC: Area under the curve mIoU: mean intersection over union, ESCC: esophageal squamous cell carcinoma, HNSCC: head and neck squamous cell carcinoma. Color legend: (i) light green: all parameters <95%, (ii) medium colored green: at least one parameter >95%, (iii) dark green: all parameters >95%.

## 4. Discussion

AI development is progressing rapidly, with the introduction of several models that can perform various tasks such as the detection and the segmentation of various malignancies such as breast, pharynx, and thyroid carcinoma [27,28,29,30], and non-tumoral specimens, among others [31,32].

LN metastases are one of the most important prognostic factors for staging malignancies [2]. The histological evaluation of nodal specimens must be performed with care and precision. However, this work, conducted manually by pathologists, is often a protracted, tedious, and possibly error-prone process that could benefit from the aid of digital pathology and AI-based algorithms designed to assist with screening LNs for metastatic disease. Indeed, the application of digital techniques to help detect LN metastases may allow pathologists to reduce turn-around times and increase their diagnostic accuracy. Several AI-based tools have been developed in the last decades for addressing this relevant issue, which are further discussed in the following sections, according to their fields of application.

**Table 2 cancers-15-02491-t002:** Characteristics of the three studies testing several types of algorithms.

Author, Year, Country	Organ	N. of WSIs *	Stain	N. of Algorithms	Scanner	Limitations
Bejnordi, 2017, Holland	BC LNS	160−, 110+/80−, 49+	H&E	32	Pannoramic 250 Flash II,3DHISTECH, NanoZoomer-XR Digital slidescanner C12000-01 Hamamatsu Photonics	No ITC
Bandi, 2018, Holland	BC LNS	899 (558 − 341+)/500	H&E	37	3D Histec P250; Philips IntelliSite Ultra Fast; Hamamatsu XR C12000	NA
Kim, 2020 South Korea	BC LNS	197/100	H&E	4	Pannoramic 250 FLASH, 3DHISTECH Ltd.	TL

* training/testing. Abbreviations: BC: breast cancer, LNS: sentinel lymph node, ITC: isolated tumors cells, H&E: hematoxylin and eosin, TL: transfer learning.

### 4.1. AI and Nodal Breast Cancer Metastases

The majority (70%) of the included studies in our systematic review focused on the evaluation of sentinel LNs in breast cancer patients. While localized breast cancer has a five-year survival rate of >95%, the presence of LN metastases drops the survival rate to 85% [33]. Based on the diameter of clusters of tumor cells, metastases can be divided into three categories: macrometastases, micrometastases, or isolated tumor cells (ITC), which reflect the “N” classification of breast cancer staging according to the eighth edition of the TNM staging criteria [2]. The biological significance of an ITC is debated and, according to the WHO, LNs just containing lTC are currently excluded from the total positive nodal count for the purposes of the N classification [33].

In 2003, Weaver et al. [19] were among the first investigators to use an automatic LN metastasis detection system on WSIs. Their system was based on a sensor capable of recognizing cells stained by IHC for cytokeratins. Those authors showed that their tool identified 19 of 20 (95.0%; 95% CI 75% to 100%) cases with micrometastases. In the only case where micrometastases were missed, the cancer cells were placed outside the physical limits of slide scanning for the instrument. It is also important to note that slides with excessive stain debris could not be analyzed by the system [19]. The use of IHC stained WSIs was reported by Clarke et al. [21] and Holten-Rossing et al. [20]. Clarke’s [21] models reached sensitivities of 57.5% for ITCs (<0.200 mm), 89.5% for micrometastases, and 100% for larger metastases, while Holten-Rossing and colleagues [20] achieved a sensitivity of 100% without any false negative. Despite the advantage of this IHC-based method, it requires longer stain times and is subject to increased complexity. However, this method could be particularly useful in specific settings such as patients that underwent neoadjuvant chemotherapy where nodal tissue may have drug-induced changes or an inflammatory/fibrotic response.

Undoubtedly, a turning point in the research of AI-based tools for detecting LN metastases was in 2016, represented by the CAMELYON16 challenge (CancerMetastases in Lymph Nodes Challenge) [25]. As a result, several improvements in computer programming resulted in AI capable of analyzing large WSI files. Several AI-based algorithms were developed for detecting metastases of various tumors. Among these was an interesting algorithm owned by Google called LYNA (LYmph Node Assistant), which highlights areas suspicious for the presence of metastatic cells. LYNA was reported in two different articles [34,35]. Steiner et al. [34] designed a fully crossed, intermodal, multi-reader study to evaluate the performance metrics for both assisted and unassisted reads. The pathologists in this study interpreted all the images in both modalities, with or without assistance, in two sessions separated by a wash-out period of at least four weeks. The results stated that all pathologists performed better than the algorithm alone with regard to both sensitivity and specificity; however, when they reviewed the images with AI assistance, the average time of review per image was significantly shorter, especially in negative and micrometastatic LNs. Of note in the study by Steiner et al. is that the researchers also examined WSI for ITC. Liu et al. [35] reached similar results with the algorithm that performed best in the CAMELEON16 challenge (slide level AUC 99.3 vs. 99.4). Relying on the LYNA, through an exhaustive screen for each slide at a high-power magnification, the authors propose the application of this AI in screening LNs highlighting ROIs. In a second phase, these areas could be evaluated by single pathologists, ignoring false positives, and interpreting only the true positive regions. Another algorithm proposed by Khalil et al. [36] took between 2.4 and 9.6 min per WSI to detect metastasis depending on the amount of the graphics processing unit (GPU) used.

As highlighted in Table 1, most of the algorithms in our review appeared to struggle with the task of detecting isolated tumor cells. In order to achieve high sensitivity for small ITCs, AI-based digital tools likely have to allow for a higher rate of false positive results. Indeed, examples of misdiagnosed tissue as micrometastasis could be hypertrophic lymphoid follicles, reactive venules or capillaries, and macrophages. Non-histological pollutants such as paraffin debris, bubbles, or stains can also be incorrectly interpreted as a metastasis by the algorithm. Such errors, however, can be addressed by incorporating oversight of the results by an experienced pathologist. For discrepant cases, additional H&E-stained sections or IHC can be performed. In the future, novel technology such as non-destructive 3D pathology may be used and coupled with the application of AI [37].

### 4.2. Public Challenges

One mechanism that has facilitated the development of AI algorithms has been through public challenges for specific tasks. Since the first challenges in 2007, the number of challenges per year has steadily been increasing [38]. As noted, a key turning point in this field occurred due to the CAMELYON16 challenge [25] where humans were compared with AI models to detect LN metastases by measuring the time required for reaching a correct diagnosis. The algorithms developed for this public challenge performed better than the 11 involved pathologists in identifying micrometastases. For example, when there was a time limit imposed for the detection of a tumor cell cluster of a diameter 0.2 to <2 mm, the AUC for the best algorithm was 0.994 (95% CI, 0.983–0.999) versus a mean AUC for the pathologists of 0.810 (range, 0.738–0.884; *p* < 0.001). Nevertheless, the AI models did not surpass the pathologists if no time restriction was applied (AUC = 0.943). A limitation of the CAMELYON16 competition was that WSI with ITCs were not provided. Further efforts were made with the CAMELYON17 challenge [26]. For this subsequent public challenge, the dataset of slides was divided into 100 artificial patients representing different pN stages for assessing the ability of the participating algorithms to perform automatic pN staging and getting closer to a real-world simulation. In general, the AI systems created were not only able to detect the presence of metastases, but were also able to measure their extent, including ITC, and hence better determine an accurate pN-stage.

However, even the best combination of algorithms only correctly classified 77% of patients at the slide level, where the best ranked team wrongly classified 13,4% of the slides in the test set. Overall, ten slides containing micrometastases and four slides containing macrometastases were missed [26], which would of course be unacceptable in clinical practice. One putative strategy to overcome such a problem would be to increase the sensitivity of these AI systems, even though this could increase the number of false positives. These early results imply that perhaps algorithms developed for research purposes to automatically detect LN metatastatic disease are not yet ready to be fully adopted in daily practice.

### 4.3. Intraoperative Consultation

Only a few studies investigated the implementation of AI to assist with frozen sections. The intraoperative evaluation of a sentinel LN is particularly demanding in this clinical setting, even for an experienced pathologist, because of artifacts such as tissue compression, nuclear ice crystals, sections with folds, and stain nuances which differ from formalin-fixed paraffin-embedded (FFPE) material, overall leading to inferior image quality [39]. Kim et al. [40,41] proposed a transfer learning to effectively train their CNN model for the identification of metastatic breast cancer cells on frozen tissue section digital slides. Transfer learning relies on the re-utilization of a model trained on one task to a second, related task by adding modifications. The authors exploited data of annotated WSIs from FFPE samples to train CNNs working on frozen sections. The best algorithms detected metastasis with an AUC of 0.805 and a processing time of 10.8 min. In conventional (human) frozen section examinations, the time between sample receipt and a rendered pathological diagnosis typically spans from 20 to 30 min, including gross specimen inspection, tissue freezing, sectioning, staining, and microscopic examination [42]. For a digital workflow, the time for scanning frozen sections may vary depending on the size, type of scanning machine, magnification, and focus z-stacking, but it generally ranges from 3 to 9 min [42,43]. Therefore, the application of an efficient AI system in the intraoperative consultation setting, despite increased diagnostic times, could be useful in particularly demanding cases, especially when a second opinion by a remotely located colleague is required. For these reasons, this technology is likely suitable for use in routine practice.

### 4.4. Tumors Other Than Breast Cancer

Our review identified seven studies where metastatic disease in LNs was from non-breast cancer series including gastric cancer (3/7), squamous carcinoma (2/7), colorectal carcinoma (1), and lung cancer. For gastric cancer metastatic LN series, Matsumoto et al. [14] combined H&E with deep UV excitation fluorescence microscopy and tested different AI models on both types of images acquired. Their results were excellent (AUC = 98.8) and these authors also demonstrated that automated analysis with fluorescence images achieved rates of detection of LN metastasis as accurately as that with H&E images. In the 222 patient cohort of Hu et al. [23], 51 were treated with neoadjuvant chemotherapy (NACT). In this particular study, these researchers demonstrated that their AI model was effective and can accordingly be confidently used for LN screening after NACT. Huang et al. obtained a slide-level [18] AUC curve of 0.9936 for gastric LN metastasis using a weakly supervised algorithm, based on ResnNet50 architecture. Moreover, their proposed method significantly enhanced the sensitivity of ITC recognition and micrometastases identification while shortening the review time per slide. In 2020, Pan et al. [44] employed an algorithm developed on a slide set of metastatic esophageal squamous cell carcinoma to screen LNs suspicious for metastases from the throat and lung. By relying on transfer learning, the applied AI tool reached an accuracy of 96.7% and 90%, respectively, for each type of cancer. Nodal metastases from head and neck squamous cell carcinoma cases were also tested by the algorithm developed by Tang et al. [24] in 2020, gaining ever higher sensitivity rates (100%), but with less specificity (75.9%). Chuang et al. [16], exploiting a ResNet50 model, built a weakly supervised algorithm that performed well in identifying both macro- and micrometastases from colorectal cancer with an AUC of 0.9993 and 0.9956, respectively, at the slide level. However, when focusing on ITC, these values dropped to 0.7828. Finally, in 2019, Pham et al. [22] employed an AI tool called HALO to recognize metastases of nearly all subtypes of lung carcinoma (excluding small cell lung cancer), achieving very high sensitivity rates but with lower specificity.

### 4.5. Limitations

One key limitation of all the herein studied AI systems is that they were designed to detect just one main pathological lesion (i.e., the detection of metastatic cells), so that they were unable to recognize other rare, but still relevant, key histological features such as co-occurring pathologies (e.g., lymphoma or infection) involving LNs. Secondly, the reported AI tools frequently faced difficulties when metastatic foci were particularly small in size (e.g., ITCs). Finally, DL-based algorithms were often expensive to develop and deploy, therefore hindering the widespread use of this technology.

## 5. Conclusions

The published data overall indicate that the application of AI in detecting LN metastasis, with due care, is feasible for routine clinical practice in the near future. Ideally, AI-based automated analysis of LNs would assist pathologists by screening these samples and thereby augment diagnostic pathology reporting and tumor staging. A high sensitivity rate is ideally required for these novel AI systems to reach this goal. Further studies are warranted to improve the performance and workflow of this promising technology, in order to validate their adoption in the routine workflow of pathology laboratories.

## Figures and Tables

**Figure 1 cancers-15-02491-f001:**
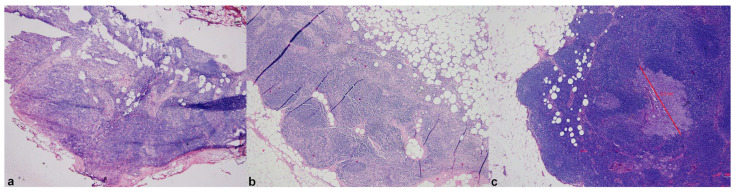
Compared WSIs of frozen (**a**) and permanent sections (**b**,**c**) of a sentinel lymph node with a micrometastasis from breast carcinoma showing up only at deeper section examination of formalin-fixed paraffin-embedded material (**c**). Reproduced with permission from Girolami I et al. [3] (original magnification ×25 (**a**) and ×50 (**b**)).

**Figure 2 cancers-15-02491-f002:**
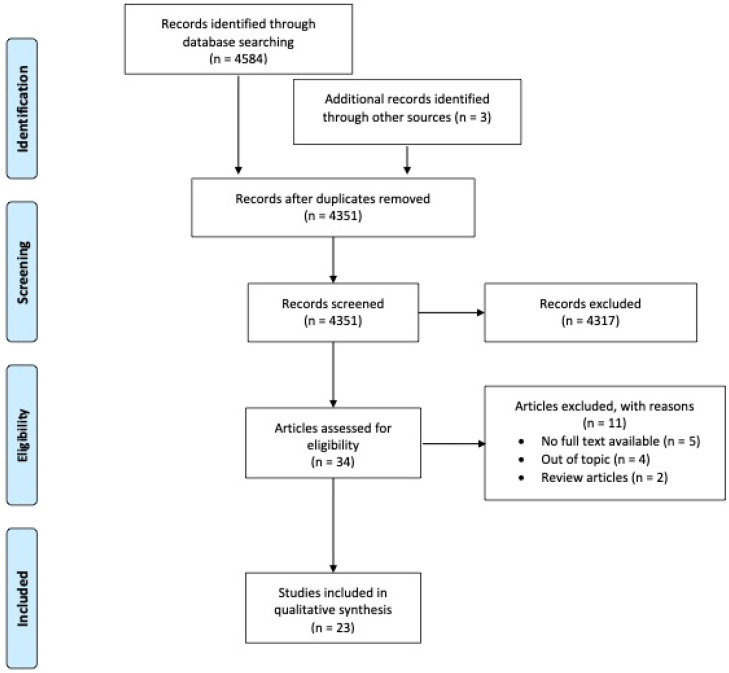
Flow diagram of the screening and exclusion articles according to PRISMA’s guidelines [11].

## Data Availability

Not applicable.

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
