# Peer review of "Value of Artificial Intelligence in Evaluating Lymph Node Metastases"

_cancers, 2023, doi:10.3390/cancers15092491_

Round 1
Reviewer 1 Report
Dear Editor,
thank you for the opportunity of reviewing this manuscript entitled “Value of artificial intelligence in evaluating lymph node metastases”.
In the present study the Authors report a systematic review regarding the application of artificial intelligence tools for supporting the recognition of lymph nodes metastases on digital slides. The work is well constructed and written and addresses a key relevant topic of routine pathology workflow. Namely, identification of nodal localizations significantly influences patients' prognosis and management. Furthermore, looking for small nodal metastases is indeed time-consuming. So, employment of automatic artificial intelligence-based systems for quick and reliable detection of metastatic cancer cells could indeed help pathologists in challenging and error-prone cases. This is particularly worth it for breast cancer, which still represents the wider proportion of examined lymph nodes for detection of metastases.
Overall, despite this manuscript appears of good quality and provides worthy insights, some minor issues need to be addressed:
· Line 65: Please capitalize the initials of food and drug administration.
· Introduction section: The final part of this section could better present the paragraphs that will later be developed in the manuscript, with regard to public challenges, Intraoperative consultation, and Tumors other than breast cancer.
· Table 1: Please provide a better legend of such visual information (e.g., different shades of green).
· Line 207: As reported for Steiner’s study, please report extensively evidence from reference [44].
· Intraoperative consultation section: It would be preferable to reformulate the conclusive part of this paragraph.
· References: Probably due to an error in the manuscript drafting, some bibliographical entries are missing. Therefore, please provide.
Minor editing of English language required.
Author Response
In the present study the Authors report a systematic review regarding the application of artificial intelligence tools for supporting the recognition of lymph nodes metastases on digital slides. The work is well constructed and written and addresses a key relevant topic of routine pathology workflow. Namely, identification of nodal localizations significantly influences patients' prognosis and management. Furthermore, looking for small nodal metastases is indeed time-consuming. So, employment of automatic artificial intelligence-based systems for quick and reliable detection of metastatic cancer cells could indeed help pathologists in challenging and error-prone cases. This is particularly worth it for breast cancer, which still represents the wider proportion of examined lymph nodes for detection of metastases. Thank you for the opportunity of reviewing this manuscript entitled “Value of artificial intelligence in evaluating lymph node metastases”.
We thank the Reviewer for appreciating our manuscript.
Overall, despite this manuscript appears of good quality and provides worthy insights, some minor issues need to be addressed:
- Line 65: Please capitalize the initials of food and drug administration.
We thank the Reviewer for the comment. Proper change has accordingly been performed.
- Introduction section: The final part of this section could better present the paragraphs that will later be developed in the manuscript, with regard to public challenges, Intraoperative consultation, and Tumors other than breast cancer.
We thank the Reviewer for this suggestion. A paragraph has been added to the final part of the Introduction section explaining the topic further discussed in the paper.
- Table 1: Please provide a better legend of such visual information (e.g., different shades of green).
We thank the Reviewer for the comment. Table 1 has been edited accordingly to the Reviewer’s suggestion.
- Line 207: As reported for Steiner’s study, please report extensively evidence from reference [44].
We thank the Reviewer for the comment. Proper considerations about the results of the mentioned study were added to the revised version of the manuscript.
- Intraoperative consultation section: It would be preferable to reformulate the conclusive part of this paragraph.
We thank the Reviewer for this recommendation. The corresponding paragraph has been checked and then reformulated.
- References: Probably due to an error in the manuscript drafting, some bibliographical entries are missing. Therefore, please provide.
We do apologize for this inconvenience. The whole manuscript has been checked and the references list properly edited.

Reviewer 2 Report
Dear Authors,
The manuscript is good to read however there are some recommendations that need to be addressed.
Minor comments:
1. In the introduction section, in line 61 authors used the term “AI” and “DL algorithm”.
Recommendation: It is recommended that authors provide a reference for line 61, in case readers wish to follow these terms.
2. Before line 63, the author could possibly mention how FDA-approved AI-enabled medical devices have been increasing in the past years overall as well as in Pathology.
Recommendation: It is recommended that the author cite this https://www.medrxiv.org/content/10.1101/2022.12.07.22283216v2.full-text paper which will provide updated status of FDA approved AI/ML landscape for broader introduction into the topic.
Major comments:
1. Please list all the MeSH terms that were used to screen the literature and also list the corresponding search platform. When search terms mentioned in this version of the manuscript were used in the suggested combination, the resulting number of studies was significantly higher. Such discrepancies must be addressed.
Minor editing of the English language and spell check required.
Author Response
The manuscript is good to read however there are some recommendations that need to be addressed.
We thank the Reviewer for appreciating our manuscript.
- In the introduction section, in line 61 authors used the term “AI” and “DL algorithm”. Recommendation: It is recommended that authors provide a reference for line 61, in case readers wish to follow these terms.
We thank the Reviewer for the comment. The corresponding phrase has been provided with a novel reference, which fully explains in detail the mentioned topics.
- Before line 63, the author could possibly mention how FDA-approved AI-enabled medical devices have been increasing in the past years overall as well as in Pathology. Recommendation: It is recommended that the author cite this https://www.medrxiv.org/content/10.1101/2022.12.07.22283216v2.full-text paper which will provide updated status of FDA approved AI/ML landscape for broader introduction into the topic.
We thank the Reviewer for suggesting such an interesting paper, which has been added to the reference list of the revised manuscript.
- Please list all the MeSH terms that were used to screen the literature and also list the corresponding search platform. When search terms mentioned in this version of the manuscript were used in the suggested combination, the resulting number of studies was significantly higher. Such discrepancies must be addressed.
We thank the Reviewer for these suggestions. Within the revised version of the manuscript, the search strategy has been further specified in the Material and Methods and fully detailed in the novel Table S1. Moreover, as WSI technologies have witnessed a significant increase only since the 2000s, we decided not to consider articles published in the previous decades.

Round 2
Reviewer 2 Report
Dear Authors,
The revised manuscript has addressed all the comments.
My overall recommendation to the editor is: Accept.
Minor.